# Surgical Management of Sporadic Peripheral Nerve Schwannomas in Adults: Indications and Outcome in a Single Center Cohort

**DOI:** 10.3390/cancers13051017

**Published:** 2021-03-01

**Authors:** Julian Zipfel, Meizer Al-Hariri, Isabel Gugel, Alexander Grimm, Volker Steger, Ruth Ladurner, Michael Krimmel, Marcos Tatagiba, Martin U. Schuhmann

**Affiliations:** 1Department of Neurosurgery, University Hospital Tuebingen, 72076 Tuebingen, Germany; meizer.alhariri@med.uni-tuebingen.de (M.A.-H.); isabel.gugel@med.uni-tuebingen.de (I.G.); marcos.tatagiba@med.uni-tuebingen.de (M.T.); martin.schuhmann@med.uni-tuebingen.de (M.U.S.); 2Department of Thoracic and Cardiovascular Surgery, University Hospital Tuebingen, 72076 Tuebingen, Germany; volker.steger@med.uni-tuebingen.de; 3Department of Neurology, University Hospital Tuebingen, 72076 Tuebingen, Germany; alexander.grimm@med.uni-tuebingen.de; 4Department of General, Visceral and Transplant Surgery, University Hospital Tuebingen, 72076 Tuebingen, Germany; ruth.ladurner@med.uni-tuebingen.de; 5Department of Maxillofacial Surgery, University Hospital Tuebingen, 72076 Tuebingen, Germany; michael.krimmel@med.uni-tuebingen.de

**Keywords:** schwannoma, peripheral nerve sheath tumors, microsurgery

## Abstract

**Simple Summary:**

Peripheral nerve sheath tumors are associated with significant morbidity. Clinical presentation, tumor location, and therapeutic strategies are variable. We aim to provide high-quality data concerning the results of interdisciplinary operative approaches for surgical resection of schwannomas. Understanding the anatomical and functional challenges of surgical interventions in the peripheral nervous system can help to enhance the outcomes of these therapies. We aim to highlight the need for interdisciplinarity and provide evidence for both excellent functional outcomes, as well as improved quality of life for patients undergoing sporadic schwannoma surgery.

**Abstract:**

Most sporadic peripheral nerve sheath tumors in adults are schwannomas. These tumors usually present with significant pain but can also cause neurological deficits. Symptomatology is diverse, and successful surgical interventions demand interdisciplinarity. We retrospectively reviewed 414 patients treated between 2006 and 2017 for peripheral nerve sheath tumors. We analyzed clinical signs, symptoms, histology, and neurological function in the cohort of adult patients with schwannomas without a neurocutaneous syndrome. In 144 patients, 147 surgical interventions were performed. Mean follow-up was 3.1 years. The indication for surgery was pain (66.0%), neurological deficits (23.8%), significant tumor growth (8.8%), and suspected malignancy (1.4%). Complete tumor resection was achieved on 136/147 occasions (92.5%). The most common location of the tumors was intraspinal (49.0%), within the cervical neurovascular bundles (19.7%), and lower extremities (10.9%). Pain and neurological deficits improved significantly (*p* ≤ 0.003) after 131/147 interventions (89.1%). One patient had a persistent decrease in motor function after surgery. Complete resection was possible in 67% of recurrent tumors, compared to 94% of primary tumors. There was a significantly lower chance of complete resection for schwannomas of the cervical neurovascular bundle as compared to other locations. The surgical outcome of sporadic schwannoma surgery within the peripheral nervous system is very favorable in experienced peripheral nerve surgery centers. Surgery is safe and effective and needs a multidisciplinary setting. Early surgical resection in adult patients with peripheral nerve sheath tumors with significant growth, pain, neurological deficit, or suspected malignancy is thus recommended.

## 1. Introduction

For neurosurgeons, peripheral nerve sheath tumors are not usually a daily occurrence. These tumors comprise a heterogenous group, consisting mostly of benign tumors (such as neurofibromas and schwannomas) as well as malignant neoplasms (malignant peripheral nerve sheath tumor, MPNST). Peripheral nerve sheath tumors can have significant morbidity, especially due to local complications [1]. Often, significant neurological impairments, like sensorimotor deficits or pain, are associated with these pathologies [2]. Most tumors are sporadic, and in the remaining cases, there is mostly an association with Neurofibromatosis Type 1 (NF1), Type 2 (NF2), or Schwannomatosis.

Benign nerve sheath tumors in NF1 are usually neurofibromas. They can grow diffusely as plexiform neurofibromas (PNF) with local infiltration of the surrounding tissue and organs, or as solitary neurofibromas related to a defined peripheral nerve. Plexiform neurofibromas are a specific manifestation, typical of and confined to NF1, and 50% of patients are affected [3].

Peripheral nerve sheath tumors in NF2 patients are almost exclusively schwannomas [4,5], although hybrid tumors exist [6].

Schwannomatosis [7] is a third genetic tumor disposition syndrome for peripheral nerve sheath tumors. It is characterized by multiple peripheral schwannomas without the presence of NF2 stigmata like bilateral vestibular schwannomas, or other intradural NF2 manifestations like meningiomas or ependymomas [2]. Schwannomatosis can be associated with SMARCB1 or LZTR1 mutations and exists in a sporadic and familial pattern. It mostly affects adults, and an age of <30 years is considered an exclusion criterion for the sporadic form [8].

Singular sporadic peripheral nerve sheath tumors not associated with neurocutaneous syndrome in adults are predominantly schwannomas [9]. Since these tumors also occur in patients with NF2 and Schwannomatosis, any occurrence of more than one peripheral nerve schwannoma mandates a work-up for those diseases. In children, on the other hand, solitary sporadic benign peripheral nerve sheath tumors are rarely described, and most published cases were neurofibromas [8,10,11].

In sporadic peripheral nerve schwannomas, local or radiating pain is one of the most important symptoms. Additionally, significant tumor growth with local complaints due to mass effects and neurological deficits are indications for surgery.

We recently published our experiences of peripheral nerve sheath tumor surgery in the pediatric population [3], where most tumors are associated with NF1, or to a much lesser extent, with NF2. Sporadic tumors not associated with neurocutaneous syndrome are rare in children and adolescents.

We now present a large cohort analysis of sporadic peripheral nerve tumor, focusing on adults that underwent operations for schwannomas. Pre- and postoperative motor and sensory function, as well as the effect on pain, were the primary focus of the outcome analysis. Furthermore, the association of the tumor location and extent of the resection was analyzed. We aim to highlight the need for interdisciplinarity and provide evidence for both excellent functional outcomes, as well as improved quality of life for patients undergoing surgery for sporadic schwannomas.

## 2. Materials and Methods

A retrospective analysis of all operated cases in our institution between 2006 and 2017 was performed. We used a hospital database search method after approval by the local ethics committee (Nr.: 026/2018BO2). In total, 144 patients with 147 performed procedures were included. Patients following surgical resection of a peripheral nerve sheath tumor (ICD codes D36.1 or Q85.0) were included. Patients without follow-up data were excluded. A control or validation cohort with incidentally found tumors that were asymptomatic and were observed was not included, since the aim of the study was to investigate symptomatic surgical cases, and not the natural history of asymptomatic tumors where the histology would be unknown.

Patient evaluation included a full medical history, general physical and detailed neurologic examination, magnetic resonance imaging, as well as peripheral nerve ultrasound. In a few cases where there was doubt as to whether the tumor was the only causative factor for the complaints of the patient, electrophysiologic preoperative investigations were performed. In the standard situation, there is no diagnostic gain from such examinations. Neurologic symptoms and signs regarding the presence of clinical stigmata of a neurocutaneous syndrome, such as NF1 and NF2, were routinely screened in our peripheral nerve surgery outpatient clinic. Preoperative biopsies were not performed.

The standard function preserving technique for benign peripheral nerve tumors, which involves intracapsular tumor resection with an entry into the tumor through a part of the capsule that does not contain any sensory or motor nerve fascicles, was performed and is described in detail in the discussion. To achieve the reported outcomes, nerve stimulators and high magnification by microscope or loupes were used intraoperatively in all cases.

The retrospective analysis included all clinical signs and symptoms, histopathology, exclusion of underlying neurocutaneous syndrome, and neurological status, such as sensory and motor function. We used the MRC (Muscle Power Assessment) scale grading system for motor function. Accordingly, sensory function was graded from 0 to 5 (no sensibility, severe hypesthesia, moderate hypesthesia, mild hypesthesia, or normal sensibility). Pain was rated on an increasing pain scale from 0-3 (no, mild, moderate, or severe pain, respectively).

Statistics were performed using SPSS Statistics 25 (IBM, Armonk, NY, USA). Continuous data were presented as a mean (±SD), whereas categorical data were shown as a count, with percentages in parentheses (*n*; %). Continuous variables were tested for equality of variances by Levene’s test. Normal distributed parametric variables with equal variances were compared using the unpaired or paired t-test and Anova. A post-hoc analysis was performed, including a Bonferroni correction. The following variables were examined as potential prognostic predictors: sex, age, previous surgery for the same tumor, tumor location, and preoperative neurological deficit. A two-sided *p*-value < 0.05 was considered statistically significant. Descriptive data are provided, including standard deviation.

## 3. Results

Over the whole investigation period, the database query for peripheral nerve sheath tumors identified 3160 procedures in 1180 patients. The inclusion criteria were not met by 766 patients with 2388 cases (e.g., no surgery, peripheral nerve surgery without tumor resection, cranial nerve surgery, mainly vestibular schwannomas). Eligible for inclusion were 414 patients with 772 cases. After further exclusion of all pediatric patients, as well as patients with neurocutaneous syndromes, 160 patients with 170 procedures for sporadic peripheral nerve sheath tumors were identified. Figure 1 shows the patient inclusion flow chart.

In total, 160 patients with sporadic tumors without association to a neurocutaneous syndrome were included. Of these patients, 144 (90%) had schwannomas. The remaining 16 patients (10%) mostly had neurofibromas. The analysis of this paper focuses on sporadic schwannomas in adult patients.

Of the 144 included patients with sporadic schwannomas, the mean age was 51 (±13) years. The sex distribution was 40.3% women (*n* = 58) and 59.7% men (*n* = 86). Overall, 156 tumors were removed in 147 surgical interventions. The mean follow-up was 3.1 years. Malignant tumors were not observed. Table 1 shows the basic patient characteristics.

### 3.1. Indication for Surgery

For two thirds of all interventions, the main reason for surgery was pain (*n* = 97, 66.0%), followed by neurological deficits (*n* = 35, 23.8%), and significant tumor growth (*n* = 13, 8.8%). Suspected malignancy was the reason for two interventions (1.4%). Of all surgical interventions, 98.0% targeted only one tumor (*n* = 144). In three cases, more tumors were resected. Operations for a recurring tumor at the same site were done on six patients (4.1%). Complete tumor resection was possible in 136/147 interventions (92.5%).

Overall, the complication rate was very low. A total of 138/147 operations had an uneventful peri- and postoperative course (93.9%). In three instances, a significant postoperative hemorrhage in the tumor bed occurred without the need of surgical revision (2.0%). In two cases, there was a CSF leak (1.4%). One wound infection was observed (0.7%).

### 3.2. Location

Almost half of the removed schwannomas were located intraspinally (*n* = 72, 49.0%). The next most common location was within the cervical neurovascular bundles (*n* = 29, 19.7%), followed by the lower extremities (*n* = 16, 10.9%). Other locations included the upper extremities (*n* = 8, 5.4%), infraclavicular brachial plexus (*n* = 6, 4.1%), lateral cervical region (*n* = 5, 3.4%), dorsal neck (*n* = 3, 2.0%), and dorsal paraspinal region (*n* = 3, 2.0%). Two cases were located in the retroperitoneum (*n* = 2, 1.4%). A single case was found in the pelvis, thoracic wall, and face. When comparing the different tumor locations, no significant differences in the occurrence of pre- or postoperative pain, motor and sensory function could be observed. In cervical schwannomas, significantly more cases were recurrent tumors than in other locations (40% vs. vascular/nerve sheath 6% *p* = 0.027, brachial plexus 0% *p* = 0.047, upper extremity 0% *p* = 0.024, lower extremity 0% *p* = 0.006, and intraspinal 0% *p* = 0.006).

### 3.3. Preoperative Symptoms and Neurological Status in Schwannomas

Preoperative neurological evaluation showed mostly good motor function. Grading was performed via an MRC scale. In 113/147 interventions, patients had preoperatively full motor function of the corresponding nerves (5/5, 76.9%), in 27 cases, minor paresis in the muscles of the affected nerve was present (4/5, 18.4%), and in 6 cases, major paresis allowing for movement against gravity was present (3/5, 4.1%). In one case, movement was only possible with gravity eliminated.

Preoperatively, sensory function was more often affected. In 35 cases, anesthesia in the area of the affected nerve was present (1/5, 23.8%), in 2 cases, major sensory impairment was present (2/5, 1.4%), in 22 mild impairment (4/5, 15.0%), and in 14 moderate hypesthesia (3/5, 9.5%). Prior to interventions, a normal sensory function was only found in 74/147 cases (5/5, 50.3%).

Pain was the major indication for surgery, being present in 87.8% of cases. It was reported prior to 90 interventions as severe pain (61.2%), in 32 cases as moderate pain (21.8%) and in 7 cases as mild pain (4.8%). Only in 18/147 interventions (12.2%) was pain not present prior to surgery.

### 3.4. Postoperative Outcome

The two leading symptoms, pain and motor/sensory deficits, improved directly after 131/147 interventions (89.1%). Neurological deficit and pain remained unchanged following 14 interventions (9.5%) and worsened on 3 occasions (2%). One patient had a postoperative paresis (3/5) after intraspinal schwannoma resection with full motor function preoperatively. Another patient had moderate postoperative hypesthesia. The paresis persisted postoperatively, and the hypesthesia recovered fully. Another patient reported intensified pain.

Surgical results were evaluated again at the last follow-up, with a mean interval of 3.1 (±0.4; 0.25-10.5) years. At the last follow-up, postoperative symptoms were still better than preoperatively in 124/147 cases (84.4%). For 20 surgical interventions, pain and sensory loss were mostly unchanged (13.6%), and in 3 cases, they were worse than preoperatively (2.0%).

### 3.5. Motor Function

At the last follow-up, full motor function (5/5) was observed after 130/147 interventions (88.4%). A minor impairment (4/5) was present in 15 cases (10.2%). Only two patients (1.4%) still had moderate paresis and could only move their limb against gravity (3/5). In total, after 123/147 cases, motor function was unchanged (83.7%). One patient had, as previously stated, a deterioration in motor function. After 23 interventions, motor deficit improved (15.6%). Figure 2 compares pre- and postoperative motor function.

### 3.6. Sensory Loss

Sensory function was unimpaired postoperatively in 81/147 cases (5/5, 55.1%). In 23 cases (15.6%), mild sensory impairment was observed (4/5), 16 cases had moderate anesthesia (3/5, 10.9%), 9 had major anesthesia (2/5, 6.1%), and 18 cases had complete anesthesia (1/5, 12.3%).

In total, after 101/147 cases, sensory function was unchanged (68.7%). Three patients had a significant and persisting deterioration of sensory function (2.0%). After 43 interventions, sensory deficit improved (29.3%). Figure 3 compares pre- and postoperative sensory function.

### 3.7. Pain

Following 78/147 interventions, no pain was reported (53.1%). After 6 interventions, severe pain persisted (4.1%). After surgery, 12 cases reported moderate pain (8.2%), and 51 mild pain (34.7%). In most patients, pain improved significantly after surgery (*n* = 119, 81.0%). In two cases, pain was worse than preoperatively. In 26 cases, pain was unchanged (17.7%). Figure 4 compares pre- and postoperative pain.

Comparing pre- and postoperative function for the whole cohort, we found significantly improved motor (4.7 ± 0.6 vs. 4.9 ± 0.4, *p* < 0.001) and sensory function (3.7 ± 1–6 vs. 4.0 ± 1.4, *p* = 0.003) and pain levels (2.3 ± 0.8 vs. 0.6 ± 0.8, *p* < 0.001). Postoperative pain levels did not differ significantly between patients with severe or mild preoperative pain (0.14 +/− 0.38 vs. 0.8 +/− 0.9, *p* = 0.053).

### 3.8. Association of Outcome to Resection Status

Complete resection was possible in 4/6 recurrent tumors (67%) compared to 94% of primary tumors (*n* = 132/141, *p* = 0.014). Both pre- and postoperative sensory function was worse in patients with recurrent tumors (pre 3.7 ± 1.6 vs. 2.2 ± 1.8, *p* = 0.022, post 4.0 ± 1.4 vs. 2.7 ± 2.0, *p* = 0.026). Motor function did not differ significantly.

Of the 11 tumors in which only incomplete resection was achieved, three cases were recurrent tumors (27.3%). Concerning the location of the incompletely resected tumors, 72.7% were in the cervical neurovascular bundle (*n* = 8), and the remaining were intraspinal (*n* = 3). Post-hoc tests revealed a significantly lower chance of complete resection in schwannomas of the cervical neurovascular bundle, as compared to other locations (76 ± 44% vs. upper extremity 100% *p* = 0.028, lower extremity 100% *p* = 0.021, and intraspinal 96 ± 20% *p* = 0.004). Figure 5 shows the comparison of tumor location with the rate of complete resection.

## 4. Discussion

Several previous studies have reported on peripheral nerve sheath tumors and their surgical treatment [1,4,5,7,8,12,13,14,15,16]. To the best of our knowledge, this retrospective analysis presents the largest series on sporadic peripheral schwannoma surgery so far.

This article aims at providing an overview of the surgical outcome and considerations for treatment, since the most favorable outcomes of peripheral schwannoma surgery shown by this study demonstrate the opportunity surgery offers to the patients. In our experience dealing with patients that have an often long presurgical history of pain, the surgical option appears to be underused, since neurologists and general physicians seem to expect more potential risks from surgery and thus do not encourage their symptomatic patients to seek consultation by a peripheral nerve surgeon early on.

### 4.1. Imaging

Ideally before, but at the latest directly after the removal of any singular peripheral nerve tumor, a surveillance ultrasound screening of all extremities and brachial plexus and neck should be performed as a diagnostic screening step to identify other, smaller and not palpable tumors of the peripheral nerves, which would be a characteristic of an underlying neurocutaneous syndrome. In the case of peripheral nerve schwannomas, the identification of further tumors would have to prompt a further diagnostic work-up to exclude an underlying NF2 or Schwannomatosis.

Ultrasound seems to be a more economical alternative to a whole-body MRI, since it is far cheaper and less time-consuming. The latter is superior concerning deeper regions (e.g., lumbar roots, spinal ganglia) and seems to facilitate the differentiation of Schwannomatosis and NF2 (Godel et al., 2018). Additionally, ultrasound is easily applicable and is able to depict several schwannoma subtypes (e.g., single fascicular, multifascicular, plexiform) by scanning larger regions of the nerve. In superficial regions and small nerves, ultrasound is superior to all other imaging tools [17]. In the hands of an experienced investigator, a whole-body screening of the major nerves of all extremities, cervical region, and brachial plexus takes approximately 15–20 minutes. Currently, the main limitation of this methodology is the lack of examiners experienced in peripheral nerve sonography following a routine scheme for a whole-body investigation. Due to the rising recognition of the value of peripheral nerve ultrasound in the last decade, this lack of availability will decrease in the next 10 years.

### 4.2. Interdisciplinarity and Surgical Technique

Peripheral nerve sheath tumors present in a vast spectrum of localizations and clinical appearances. Interdisciplinary teams are not only best prepared to manage the known multiple aspects of neurocutaneous syndromes [18] but also to address peripheral nerve tumors not associated with a neurocutaneous syndrome and all possible manifestations in the different body regions.

Expertise in peripheral nerve tumor surgery and function-preserving resection techniques is warranted to manage those cases well from a surgical point of view. The standard function-preserving technique for benign peripheral nerve tumors, which make up the vast majority, as this series shows, is an intracapsular tumor resection. The usual steps are, firstly, an exposition of the tumor using the gentlest approach to the pathology with regard to approach-related pain and morbidity. This is of special importance regarding retro- or intraperitoneal and intrathoracic manifestation, where the expertise and routine of abdominal and thoracic surgeons is of the greatest importance. Laparoscopic and thoracoscopic approaches have to be considered and are suitable for smaller schwannomas. After exposition, the tumor-bearing nerve proximal and distal of the schwannoma should be identified, and if it is dissectible without additional risks (e.g., due to proximity of vulnerable veins), we recommend a circumferential looping with rubber bands. After peeling the outer connective tissue or fat layers, the tumor surface should be exposed and closely inspected to identify the intracapsular course of the function-bearing fascicles. For this purpose, higher magnification by a microscope or loupe is strongly recommended,

Bare-eye approaches carry a much higher risk. The favorable results in this series have been reached by an exclusively microsurgical approach, with a minimal magnification of 3x with loupes, or higher when using a microscope. With the help of nerve stimulators, motor function-bearing fascicles can be distinguished from purely sensory/pain fascicles. The entry through the tumor capsule is executed in the area with the largest distance to the identified fascicles. For this purpose, tumors can often be quite effectively rotated around their longitudinal axis. After the incision of the capsule, the plane of dissection needs to be identified. In schwannomas (as in neurofibromas), this can often be recognized by the flow of small amounts of a serous fluid that tends to surround approximately 50% of tumors. Larger tumors often need to be decompressed by internal tumor resection, also depending on the space available for manipulation, which is defined by the proximity to vulnerable structures like the spinal cord, veins, or arteries, before the tumor capsule can be mobilized away from the tumor, or vice versa in the case of an approach to a large tumor through a narrow opening in the capsule. Frequently, at the end of this separation of capsule and tumor, or at an earlier time, the tumor-bearing single fascicle becomes visible and needs to be removed. Since most schwannomas originate from a sensory fiber, permanent motor deficits due to the removal of the tumor-bearing fascicles are rare, and new sensory deficits, as this series shows, only occur in a minority of cases.

It is impossible to distinguish in such a retrospective analysis if persistent deficits have been created by additional injury of preoperatively not affected fascicles, by fibers during surgery, or by tumor-induced damage that could not recover. After the tumor removal, meticulous hemostasis within the capsule is achieved either by using very low power bipolar coagulation or, in case of venous bleeding, prolonged compression to induce spontaneous coagulation. A drain is only left for possible bleeding alongside the surgical approach.

### 4.3. Outcome

In most cases, preoperative motor function was good. Applying the above-mentioned principles of peripheral nerve tumor surgery and microsurgery had no significant negative impacts on postoperative motor function. This is an important result, since patients and referring doctors can be reassured that new motor deficits after surgery are a rarity that need not be feared.

Directly after surgery, the preoperative sensory deficits were still present in a large number of patients. In the follow-up analysis, this was not a significant issue anymore. Due to the retrospective character of this study, a clear distinction between full objective recovery and subjective improvement of symptoms with consequent underreporting cannot be made with absolute certainty.

Concerning pain, surgery was able to significantly improve pain for affected patients: the overall outcome was excellent. The level of preoperative pain did not correlate significantly with the postoperative pain level and is thus not a prognostic marker.

In this series, surgical resection of tumors of the cervical neurovascular bundle was significantly less complete than for other locations. This was mostly due to the involvement of cranial nerves (CN X with a recurrent laryngeal nerve, CN IX, and XII). With the benign biology of schwannomas, the threshold to be less aggressive and not damage the functionally important nerves in case of strong adherence and very difficult dissectability of tumor and capsule, as we have experienced in this location in a number of cases, is low. An incomplete resection has always been preferred over functional deficits, like swallowing difficulties or vocal cord paresis.

## 5. Conclusions

This large retrospective series of peripheral schwannoma resections demonstrates that surgery is a safe and effective treatment. Patients benefit from early surgical resection of peripheral nerve sheath tumors that cause significant pain, and sensory or motor deficits. The effects on pain are immediate and long-lasting for 80% of patients, sensory deficits improve in about 30% of cases, and motor deficits improve in about 15% of cases. Since new deficits are extremely rare, and no other major complications have been observed, patients and referring doctors can be assured of minimal surgery-associated risks. Therefore, surgical intervention is the method of choice.

## Figures and Tables

**Figure 1 cancers-13-01017-f001:**
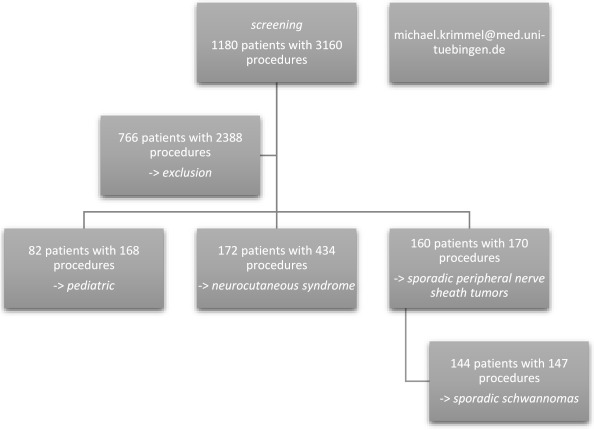
Study flow chart.

**Figure 2 cancers-13-01017-f002:**
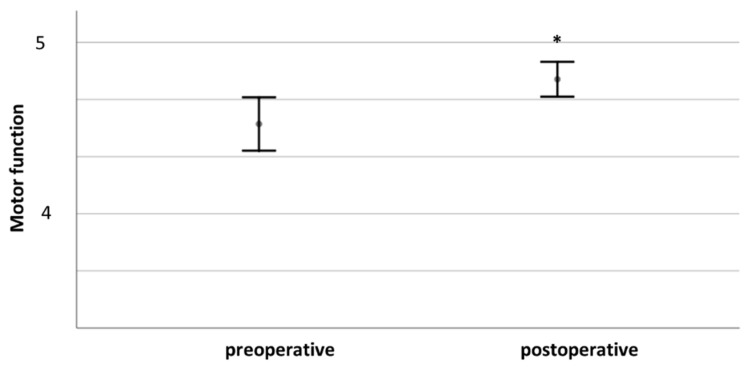
Comparison of pre- and postoperative motor function. * indicates *p* ≤ 0.05.

**Figure 3 cancers-13-01017-f003:**
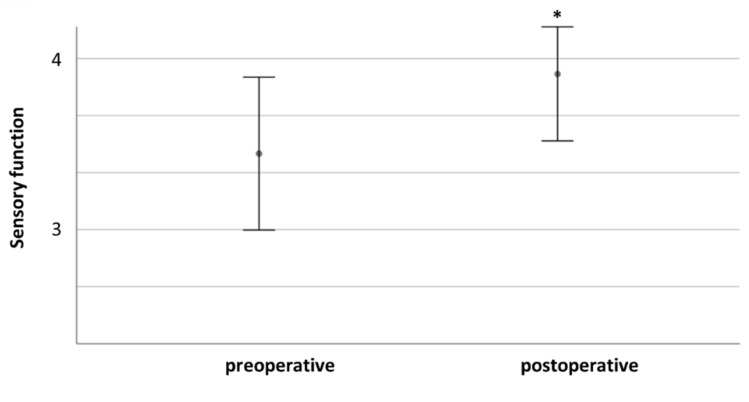
Comparison of pre- and postoperative sensory function. * indicates *p* ≤ 0.05.

**Figure 4 cancers-13-01017-f004:**
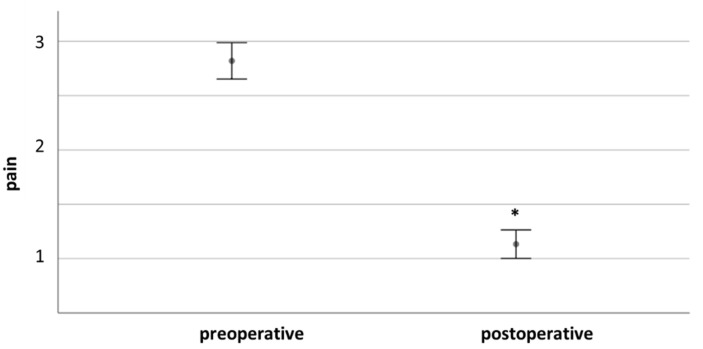
Comparison of pre- and postoperative pain. * indicates *p* ≤ 0.05.

**Figure 5 cancers-13-01017-f005:**
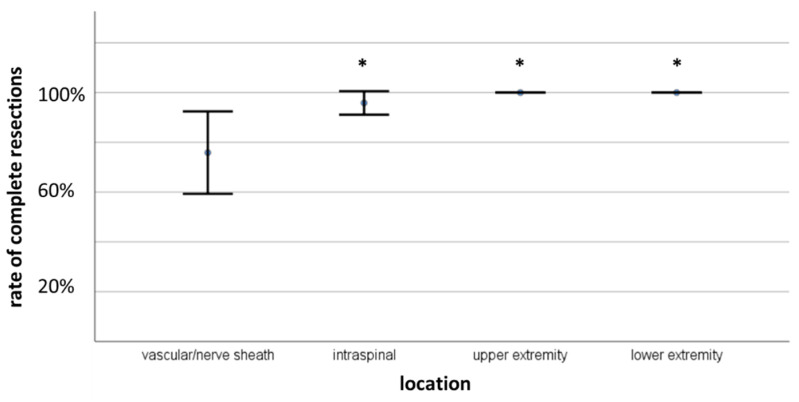
Comparison of rate of complete resection in different tumor locations. * indicates *p* ≤ 0.05.

**Table 1 cancers-13-01017-t001:** Basic patient characteristics.

Parameters	All Procedures (*n* = 147)						
Age	Mean Patient Age 51 (±13) Years						
Sex	40.3% Women (*n* = 58/144 Patients)			59.7% Men (*n* = 86/144 Patients)			
Location	49% intraspinal (*n* = 72)	19.7% neurovascular bundle (*n* = 29)	10.9% lower extremities (*n* = 16)	5.4% upper extremities (*n* = 8)	4.1% infraclavicular brachial plexus (*n* = 6)	3.4% lateral cervical (*n* = 5)	7.5% other (*n* = 11)
Indication for surgery	66.0% pain (*n* = 97)	23.8% neurological deficits (*n* = 35)		8.8% tumor growth (*n* = 13)		1.4% suspected malignancy (*n* = 2)	
Number of tumors	98% solitary (*n* = 144)			2% multiple tumors (*n* = 3)			
Surgery for recurrent tumor	4.1% yes (*n* = 6)			95.9% no (*n* = 141)			
Complete resection (CR)	92.5% CR (*n* = 136)			7.5% no CR (*n* = 11)			
Complications	93.9% no complication (*n* = 138)	2.0% local hemorrhage (*n* = 3)		1.4% CSF leak (*n* = 2)		1 wound infection	

## Data Availability

The data presented in this study are available on request from the corresponding author. The data are not publicly available due to restrictions by the ethical approval.

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
