# Peer review of "Surgical Management of Sporadic Peripheral Nerve Schwannomas in Adults: Indications and Outcome in a Single Center Cohort"

_cancers, 2021, doi:10.3390/cancers13051017_

Round 1

Reviewer 1 Report

initially I was somewhat hesitant to see intraspinal and peripheral schwannomas lumped together.

however, there are good arguments for it and i can follow the authors´train of thought.

they have a vast experience in this field and the series of sporadic schwannomas is the largest published has meaningful implications for non experts.

i have only one suggestion: the recommendation and notion about radio surgery should be left out of the conclusions, it is a (well founded) opinion, but not borne by data presented

(notwithstanding the I totally agree)

Author Response

We have removed the mentioned paragraph (lines 351-357). The reviewer’s argument is comprehensible as we have not presented data on radiosurgery and the lines mainly reflect our professional opinion and are not based on our research.

Reviewer 2 Report

The authors report a large study on sporadic schwannoma treated surgically in a single center academic setting. This is a large case series without a control group dwelling on various outcomes. The overall aim of this work, research question and discussion are not sharp enough to satisfy the readership of this journal. From a clinical science point of view the methods section is too short and inaccurate + contains results. The results section is missing a study flow chart (screening/inclusion/exclusion) as well as a table on basic patient characteristics. In general, neurocutaeous syndrome should be preferred over the rather antiquated term "phacomatosis". Overall, the manuscript would benefit from a thorough revision and adaptation to a standardized reporting protocol. I am sure that with a dedicated research question (e.g. respectability versus location), a study size this large most probably will provide exciting and valuable evidence to many practitioners in the field. 

Author Response

The authors report a large study on sporadic schwannoma treated surgically in a single center academic setting. This is a large case series without a control group dwelling on various outcomes.

Indeed, no control group was available in our analysis. In our clinical practice as surgeons of different departments, patients usually present with symptomatic tumors with a consecutive indication for surgery. Incidental tumors not associated to neurological deficits or pain are usually monitored via imaging and resection is recommended in the case of significant tumor growth. Thus, a “conservative” or “no surgery” control group is not available. The paragraph of the discussion concerning radiosurgery has been removed due to reviewer request. In our clinical experience and collaboration with other departments and hospitals, radiosurgery for peripheral nerve sheath tumors is not performed on a regular basis. No radiosurgery control group is available in our institution or collaborative partner site.

The overall aim of this work, research question and discussion are not sharp enough to satisfy the readership of this journal.

We tried to provide a sharper aim of our work with emphasizing the need for interdisciplinarity and provide data on the impact of our treatments on patients’ outcome. The analysis of tumor location and resectability was added to the research question as it is a major topic of our study yet wasn’t mentioned in the aim of our work (added lines 78-81).

From a clinical science point of view the methods section is too short and inaccurate + contains results.

We tried to improve the methods section by moving parts of it to results (lines 117-124). A thorough description of patient inclusion (86-95) and a paragraph on surgical technique were added (96-106).

The results section is missing a study flow chart (screening/inclusion/exclusion) as well as a table on basic patient characteristics.

A study flow chart was added to facilitate the readers’ comprehension of our methods. Furthermore, a basic patients characteristics table was added, repeating the information in the text.

In general, neurocutaeous syndrome should be preferred over the rather antiquated term "phacomatosis".

The term phacomatosis is used in our daily practice, while admittedly being outdated. We changed the text accordingly and replaced it with neurocutaneous syndrome (lines 24, 63, 74, 108, 114, 267, 286, 287)

Overall, the manuscript would benefit from a thorough revision and adaptation to a standardized reporting protocol. I am sure that with a dedicated research question (e.g. respectability versus location), a study size this large most probably will provide exciting and valuable evidence to many practitioners in the field. 

We hope to have thoroughly addressed the reviewer’s criticism.

Reviewer 3 Report

In this retrospective study the authors reviewed their experience between 2006 and 2017 on the management of adults schwannomas, especially analyzing pre- and postoperative motor and sensory function as well as the effect on pain.

The paper is well documented, scientifically satisfactory and rigorous. The surgical case series is very wide and varied and the conclusions are correct and shareable. The bibliography is good.

Author Response

Apparently, no specific improvements were demanded by the reviewer.

Round 2

Reviewer 2 Report

Happy with the changes made. I assume the comment on author contribution is of in-house nature.